# A Polymer Thick Film on an Organic Substrate Grid Electrode and an Open-Source Recording System for UHF MRI: An Imaging Study

**DOI:** 10.3390/s24165214

**Published:** 2024-08-12

**Authors:** Yinching Iris Chen, Ilknur Ay, Francesca Marturano, Peter Fuller, Hernan Millan, Giorgio Bonmassar

**Affiliations:** 1AA. Martinos Center Massachusetts General Hospital, Harvard Medical School, Charlestown, MA 02129, USA; ychen16@mgh.harvard.edu (Y.I.C.); iay@mgh.harvard.edu (I.A.); fmarturano@mgh.harvard.edu (F.M.); hernan.millan@mgh.harvard.edu (H.M.); 2PLF Consulting, 258 Harvard Street #324, Brookline, MA 02446-2904, USA; peterlfuller@gmail.com

**Keywords:** ECoG, artifacts, magnetic resonance imaging, rats, 4.7 teslas, 9.4 teslas, MRI heating, open-access software, recording system

## Abstract

Electrocorticography (ECoG) is a critical tool in preclinical neuroscience research for studying global network activity. However, integrating ECoG with functional magnetic resonance imaging (fMRI) has posed challenges, due to metal electrode interference with imaging quality and heating around the metallic electrodes. Here, we introduce recent advancements in ECoG grid development that utilize a polymer-thick film on an organic substrate (PTFOS). PTFOS offers notable advantages over traditional ECoG grids. Firstly, it significantly reduces imaging artifacts, ensuring minimal interference with MR image quality when overlaying brain tissue with PTFOS grids. Secondly, during a 30-min fMRI acquisition, the temperature increase associated with PTFOS grids is remarkably low, measuring only 0.4 °C. These findings suggest that utilizing ECoG with PTFOS grids has the potential to enhance the safety and efficacy of neurosurgical procedures. By providing clearer imaging results and mitigating risk factors such as excessive heating during MRI scans, PTFOS-based ECoG grids represent a promising advancement in neurosurgical technology. Furthermore, we describe a cutting-edge open-source system designed for simultaneous electrophysiology and fMRI. This system stands out due to its exceptionally low input noise levels (<0.6 V peak-to-peak), robust electromagnetic compatibility (it is suitable for use in MRI environments up to 9.4 teslas), and the inclusion of user-programmable real-time signal-processing capabilities. The open-platform software is a key feature, enabling researchers to swiftly implement and customize real-time signal-processing algorithms to meet specific experimental needs. This innovative system has been successfully utilized in several rodent EEG/fMRI studies, particularly at magnetic field strengths of 4.7 and 9.4 teslas, focusing on the somatosensory system. These studies have allowed for detailed observation of neural activity and responses within this sensory system, providing insights that are critical for advancing our understanding of neurophysiological processes. The versatility and high performance of our system make it an invaluable tool for researchers aiming to integrate and analyze complex datasets from advanced imaging and electrophysiological recordings, ultimately enhancing the depth and scope of neuroscience research.

## 1. Introduction

Preclinical models play a crucial role in advancing our understanding of the human brain, with brain imaging in small animals offering insights into the effects of disease, injury, genetics, drugs, and treatments on brain function. By integrating functional magnetic resonance imaging (fMRI) with an intracranial electroencephalogram (iEEG) or electrocorticogram (ECoG) via multimodal imaging, researchers can gain a comprehensive view of the functional connectivity and the intrinsic oscillatory behavior of cortical regions. Despite the clear advantages, small animal research faces challenges due to the need for more sophisticated tools for high-quality ECoG-fMRI recordings. Combining ECoG with functional magnetic resonance imaging (fMRI) presents a significant challenge, primarily due to the interference caused by metal electrodes. Substantial metal artifacts in MR images, especially in ultra-high-field (UHF) MRI, may obscure a large volume of the brain tissue around the electrodes (Figure 1). The main reason for this distortion is the susceptibility mismatch caused by the difference in magnetic properties between the electrodes and the surrounding water or tissues [1].

These artifacts, or ‘blind spots’, are especially problematic in fMRI with echo-planar imaging (EPI), which is particularly sensitive to magnetic susceptibility. While these artifacts might not hinder the identification of functional activity in brain areas far away from the electrodes, they significantly obstruct the mapping of local responses near the electrode site. This leads to incomplete and biased activation pattern mapping, particularly affecting those areas near the electrode tracks. This study focuses on developing ECoG grids using polymer-thick film (PTF) deposited on an organic substrate (PTFOS, Figure 2a). This approach aims to ensure safety during MR imaging and produce artifact-free images [2]. Previous research has shown that PTF-based EEG caps can provide artifact-free MR imaging and high-quality EEG recording, setting a precedent for applying this technology in ECoG [3,4,5]. The proposed design includes thin, flexible grids that conform to the brain’s surface, potentially reducing post-surgical complications and improving recording placement. The PTFOS grids underwent testing to evaluate MRI heating and the presence of imaging artifacts. The development of PTFOS grids represents a significant advancement in ECoG technology, addressing previous limitations and offering a safer, more effective tool for preclinical research. This research builds upon our earlier findings, published in *Radiology* [2], demonstrating that the PTFOS technology remains artifact-free even at higher MRI fields, specifically at 9.4 teslas (T) compared to the 7 T previously studied. Moreover, while our earlier research focused solely on post-mortem imaging [2], this study extends our scope by examining MR images of live rats with implanted PTFOS devices, using various MRI techniques. The focus of this manuscript is to demonstrate that the new PTFOS implantation provides usable MR images, in contrast to the distorted images produced by traditional ECoG devices that lead to disturbances in magnetic field homogeneity. We tested image quality with PTFOS implantation using various clinical MRI sequences with different sensitivities to magnetic field homogeneity, including T1-weighted (T1W), proton density (PD), T2-weighted (T2W), fast low-angle shot (FLASH), ultra-short TE (UTE), diffusion tensor imaging (DTI), and functional MRI (fMRI). The conventional spin-echo-based sequences, including T1W, PD, and T2W, are typically insensitive to magnetic field inhomogeneity, whereas the FLASH sequence, a steady-state conventional sequence, as well as EPI and DTI sequences, are typically sensitive to field inhomogeneity [6].

## 2. Materials and Methods

### 2.1. PTFOS Construction

The PTFOS grid is composed of a PTF construct printed on top of an organic substrate (Figure 2b). The PTF construct is a three-layered circuit, with two insulator layers on the top and bottom (white blocks in Figure 2b) and a middle layer containing conductive Ag nanoparticles in conductive ink (i.e., the electrode, the yellow block in Figure 2b). This three-layered construct is then deposited on Gelfilm^®^ (Pharmacia and Upjohn, a division of Pfizer, New York City, NJ, USA), which is an organic substrate made of denatured collagen. Gelfilm^®^ is commonly used in neurosurgical procedures as a dura patch. The PTFOS was designed and built for the small animal (rat) cortex, to obtain full coverage from 32 electrodes of 500 μm in diameter and traces of 127 μm in width.

### 2.2. The HF-2 System 

We designed a new system (from now on, this will be referred to as HF-2 (Figure 3)), which can be used under a high-field MRI up to 9.4 T. Amplification is performed by the preamplification board (Figure 4a), while anti-aliasing filtering, level shifting, and analog-to-digital conversions are performed by the analog board (Figure 4b). The analog board unit, although non-ferromagnetic, can be positioned within the MR scanner’s bore. However, it is typically situated outside it to reduce the pickup of magnetic gradient artifacts. The digitally sampled data from the analog board is then transferred to a laptop via the CPU board (Figure 4d) and the Ethernet connection. The system operates at a sampling frequency of 13.562 kHz and is powered by two 12 V lead–acid batteries (Figure 3); in this way, it is isolated from the power line.

The HF-2 hardware has been carefully designed to minimize interactions with the static B_0_ magnetic, gradient magnetic, and radiofrequency fields. All components in the HF-2 are non-ferromagnetic, including aerospace-grade connectors made from a non-ferrous stainless-steel chassis constructed from aluminum. All metals, including the screws, connectors, and integrated circuits (ICs), have been screened for ferromagnetism using high-field neodymium magnets and the MRI static field itself. Susceptibility testing of the ICs is required because the metallurgy of ICs is rarely specified in the data sheets. Interactions with the gradient magnetic field are minimized by minimizing the size of the loops formed on the printed circuit board (PCB) traces. Radiofrequency (RF) interference is controlled by using three approaches: (1) PCB layouts designed with 50 Ω impedance traces to minimize RF emissions and pickup, (2) double-shielding of the cables and chassis, and (3) capacitive filter connectors to minimize RF leaks through apertures in the shielded chassis. 

(i) Preamplification Board: The signal-to-noise ratio (SNR) within the MRI scanner is improved when the external amplifier unit is positioned adjacent to the electrodes (i.e., at the head stage or pre-amplifier board). This setup allows the signal to be pre-amplified before potential MRI-related noise sources, which are primarily produced by the gradient and static magnetic fields (e.g., gradient switching vibrations from the helium pump), are picked up. The pre-amplifier unit is designed with a compact circuit layout to minimize the pickup of gradient and static magnetic noise. It uses the Analog Devices AD620B instrumentation amplifier (Analog Devices, Norwood, MA, USA). The AD620B features low-gain nonlinearity (10 ppm), high bandwidth (120 kHz with a gain of 100), and a wide power supply range (Vs = ±18 V) and output range (within 1.2 V of Vs) to accommodate the potentially large gradient artifacts encountered in the electrophysiological recordings taken during an MRI scan. The gain (G) on the external amplifier unit is set during board construction, depending on the dynamic range required. In situations with elevated background noise levels, such as those caused by the helium pump or high slew-rate gradients, or when conventional ECoG/depth electrodes are employed, the gain is set to G = 50 to provide a wide input range of −176 to +174 mV. If background noise levels are lower, for instance, when recording outside the MRI environment, a higher gain setting of G = 500 is used, providing an input range of −17.6 to +17.4 mV.

(ii) Analog Board: The ADS1254E is a low-power, low-noise, high-speed ADC with 24-bit resolution that provides the high dynamic range required for electrophysiological recordings during an MRI scan (Texas Instruments, Dallas, TX, USA). Each ADS1254 chip has four differential inputs, which can be multiplexed at up to 1 kHz per channel. Alternatively, a single channel can be sampled at up to 20 kHz. To provide 32 acquisition channels, an architecture using 8 ADS1254s was chosen, each running with 4-channel multiplexing at 1 kHz. If higher-rate sampling is required, a single channel from each ADS1254 can be sampled, resulting in 8 channels at 20 kS/s per channel. The sampling scheme can be selected at a high level in the data acquisition software (LabVIEW 2020.0.1, National Instruments, Austin, TX, USA) or at a low level by interacting directly with the PCB board (see below). The analog input to each ADS1254 is first filtered for anti-aliasing and then re-scaled to a 0–5 V dynamic range, using an op-amp-based scaling and level-shift circuit. Anti-aliasing filtering is achieved using a one-pole low-pass RC divider with a software-selectable cut-off frequency, yielding a DC bandwidth of 300 Hz or 4 kHz. The selection is performed by a PCB board-controlled transistor switch that controls the capacitance of the RC circuit, which can be controlled with HF-2’s LabVIEW data acquisition software. If necessary, these cut-off frequencies can be adjusted during board construction by changing the component values of the RC divider. In addition, the ADS1254 has its own anti-aliasing filter from DC to 4.24 kHz. The ADS1254 provides a 19.1-bit RMS effective resolution value due to inherent noise within the integrated circuit, which provides an effective resolution of 0.71 µV in the least significant bit (LSB) when the gain is set to G = 50, and an effective resolution of 0.071 µV LSB when the gain is set to G = 500. Noise measurements were performed in a shielded room (Braden Shielding Systems, Tulsa, OK, USA) with all amplifier inputs short-circuited to ground, yielding a peak-to-peak noise of 6 µV when the gain was set to G = 50 and a peak-to-peak noise of 0.6 µV when the gain was set to G = 500. For the recordings described in Section 2.3 and Section 3, a gain of G = 50 was used.

(iii) CPU Board: The CPU board (Figure 3 and Figure 4d) is engineered to bridge the analog and clock boards with a computer board (CompactRIO, National Instrument) to enhance integration and performance. The CPU board has an embedded FPGA controller and a real-time processor with NI Linux Real-Time and integrated Gigabit Ethernet, which is used to communicate and transfer data to the Windows computer/laptop. Auxiliary digital input signals, like triggers, are acquired by the CPU board and are accessible from the front panel, synchronizing data acquisition with the MRI scanner’s TTL pulse and signaling the start of each fMRI volume acquisition. This signal synchronizes data acquisition with the MRI scanner’s TR or trigger pulse, a TTL pulse that signals the start of each volume acquisition in fMRI. The HF-2 is synchronized with the MRI acquisition system through the analog board synchronization, which was implemented as follows. 

(iv) Clock board: The clock board features the Novatech DDS8m, a versatile device that is capable of generating cosine, sine, and ACMOS/TTL outputs simultaneously. Its frequency generation capabilities are finely tunable in 1 µHz increments within a broad range from 100 Hz to 100 MHz. Furthermore, direct digital synthesis (DDS8m) supports programmable frequency shift keying (FSK) and CHIRP outputs, making it adaptable for various applications. Control over the device is achieved through an RS232 serial interface, allowing for precise adjustments. The phase can be adjusted with 14-bit resolution, offering high precision in terms of output waveforms. Frequency accuracy is maintained at 1 ppm, which is attributable to the onboard temperature-compensated crystal oscillator (TCXO) or an optional external time-based device, ensuring reliable performance. Powering the DDS8m requires a 5 VDC source, aligning with standard power supplies, thereby facilitating its integration into broader systems. The MRI scanner’s 10 MHz master clock is connected to the clock board (Figure 4 and Figure 5), based on a DDS device. The clock board changes the 8 MHz to a new clock based on the device’s sampling rate specified in Table III of the ADS 1254 datasheet. The clock is then fed into the FPGA for buffering and distributed to the analog board’s 8 A/D chips (i.e., ADS 1254).

### 2.3. MRI Testing

To investigate the heating profile of the PTFOS during typical MRI sessions, we made a brain phantom with MR thermodynamic properties similar to those of rat brain tissues [8]. The PTFOS grid was placed over the phantom, and the temperature changes of the phantom were monitored during a 30-min fMRI session, with repetitive EPI acquisitions on a 4.7 T Brucker Biospin scanner. We used a Bruker volume coil for transmission and a 4-channel Bruker phased array surface coil for receiving. fMRI was obtained using a repetitive single-shot EPI sequence with the following parameters: average 1, TE 12.76 ms, TR 1000 ms (temporal resolution: 1 s per time point), field of view (FOV) 35 × 35 mm, dimension 82 × 82 mm, spatial resolution 0.427 × 0.427 mm with 1 mm slice thickness, number of slices equal to 25, and sweep width of 250 kHz. Temperatures at the core and superficial areas of the phantom near the PTFOS grid electrodes were measured using eight optical sensors (OSENSA Innovations, Coquitlam, BC, Canada), which had a calibrated accuracy of 0.1 °C (PRB-400 Fiber Optic Temperature Probe, OSENSA Innovations, Burnaby, BC, Canada).

### 2.4. In Vivo Studies

MRIs were performed on adult Wistar rats (280–370 g), with either a PTFOS grid (n = 2) or Gelfilm (control; n = 1) implanted over the parietal cortex. During the implantation surgery, the rats were anesthetized (isoflurane, 4–5% for induction and 1–2% for maintenance in a medical air and oxygen mixture), placed in a stereotaxic apparatus, and underwent parietal craniotomies. To examine the impact of the connector on the imaging quality, we kept one rat with the PTFOS connector in place and the other rat without the connector. The connector was a female nickel-free custom-made Omnetics connector, compatible with MRI. It featured 32 channels, 36 pins, 0.025″ pitch, and 4 guide posts, as well as a header contact male with a pin pitch mating of 0.025″ (0.64 mm). The number of positions loaded was 36, with a row spacing of 0.030″ (0.76 mm), a style board to a cable, shrouded with 4 walls, surface mount, right angle, solder termination, a push-pull fastening and insulation height of 0.070″ (1.78 mm) with a circular contact shape, and a gold contact with underlying beryllium copper alloy and thermoplastic insulation material. Additionally, a 2.5 × 5 cm sheet of Gelfilm^®^ without a PTFOS grid or connector was implanted over the cortex in a rat, serving as our sham control. Gelfilm^®^ was used as an FDA-approved product for neurosurgery. After securing the PTFOS/Gelfilm, the skin was closed, and the animals were transferred to an MR cradle. We imaged the rats in a 9.4 T scanner (Biospin, Brucker, Ettlingen, Germany) with sequences similar to those commonly used in 3 T clinical studies, including T1-weighted (T1W), T2-weighted (T2W), proton-density (PD), FLASH, UTE 2D, diffusion tensor, and EPI for fMRI (see Table 1 for details) sequences. All imaging was performed with a Bruker volume coil for transmission and a 4-channel Bruker phased array surface coil for receiving.

### 2.5. The HF-2 Software

The main subsystems of the HF-2 software, which are written in LabVIEW (National Instruments, Austin, TX, USA), are: (a) FPGA firmware for low-level data acquisition. (b) Real-Time (RT) software for data acquisition control and data serving via an Ethernet interface. (c) Host, Windows-based data acquisition software for connecting to the RT via TCP/IP over the Ethernet interface. 

#### 2.5.1. Field-Programmable Gate Array (FPGA) Firmware

The FPGA firmware provides low-level control for synchronizing and performing analog-to-digital conversions (ADCs) using an internal or external clock. Four banks of eight channels can be connected to the ADCs. The clock and bank selections are sent to the FPGA via the Real-Time (RT) software. In addition, the firmware allows monitoring of an external sync pulse line (Sync). The FPGA assembles the data into scans or packets, along with identifiers, a synced line state (high/low), and a count of the pulses detected. These scans are transferred using a DMA FIFO to the RT software. The DMA FIFO accepts an array (length: 16) of integer (32-bit) values. The length selection is the smallest for the number of data values that must be transferred via the DMA FIFO. 

#### 2.5.2. Real-Time (RT) Software

The RT software is responsible for the following functions: TCP/IP, RS-232, and FPGA (Figure 5). The RT software at startup continuously monitors for a TCP/IP connection on port 6310. Upon establishing a connection, it monitors the connection for command messages from the Host. When requested via the TCP/IP command, the DMA FIFO is enabled, and the data are read in blocks of 200 scans and transferred to the TCP/IP communications, where payload packets of either 600 scans or 1400 scans are collected. The packet size is adjusted for performance, while the selected sample rate determines the selection. 

The RS-232 communication, upon power-up, connects to the Novatech and sets the default clock frequency for the FPGA sample rate of 12,800 samples per second. The sample rate options are 1000, 4800, 6400, 8000, 9600, and 12,800 samples per second.

The control of the FPGA firmware handles setting the bank number to acquire, resetting the sync counter, retrieving status information, and turning on/off the DMA FIFO transfer. When the DMA FIFO is on, the data are read and transferred to the TCP/IP communication to send on to the Host. 

#### 2.5.3. Host Software

The HF-2 Host software (LabVIEW 2020.0.1, National Instruments, Austin, TX, USA) is executed on a Windows laptop, allowing the operator to connect/disconnect the RT software. The software comprises three modules: the user interface, TCP/IP, and data display. Once connected, the operator can select the bank of channels, set the sample rate, monitor data from the RT, and save the data in a file for analysis. Upon launch, the Host software waits for the operator to press the Connect button to connect the TCP/IP to the data acquisition unit. The sample rate and bank number should be selected before clicking the Monitor data button. The Host software allows for the monitoring and storage of data on the disk. The data shown in Figure 6a was collected in real time from a rat during forepaw stimulation. The data consists of three files: a “.bin” file containing the ADC data, a “.trg” file containing the Sync pulse, and a “.JSON” file containing the parameter information.

#### 2.5.4. Windows LabVIEW

Figure 6 shows the Windows LabVIEW layout of the graphical user interface (GUI). The major software acquisition components are shown in Figure 6a: (A) display properties (e.g., change the color, visibility, etc.), (B) the number of misread packets, (C) N/A (debugging), (D) queue size, (E) the number of trigger/synch pulses from MRI, (F) the CLK rate versus the data output rate (see the ADS1252 specification sheet), (G) sampling rate, (H) bank of 8 channels for selection, (I) selection tab, (J) HF-2 connection/disconnection, (K) start/stop acquisition, (L) start/stop saving data, (M) exit from LabVIEW, (N) the 100-μV scale, (o) MRI trigger/synch plot, (P) the same as A, (Q) the sweep line, and (R) the 1-s mark line. 

The review components of the recorded signal are shown in Figure 6b: (A) file open/close, (B) sampling rate, (c) read backward, (D) number of displayed time points, (E) read forward, (F) end of file, (G) scale, (H) scale apply, (I) display properties (e.g., change the color, visibility, etc.), (J) indicator/pointer of the start location of the displayed data in the file, (K) synch plot, and (L) total recording time (seconds). 

Finally, the status components of the recorded signal are shown in Figure 6c: (A) read internal HF-2 status, (B) Zynq 7020 (Figure 3) DMA status, (C) 32- or 8-channel acquisition mode (32 channels = True), (D) bank selection (1–4) when in 8-channel mode, (E) the bank number selected, (F) Clk Rate (see F in Figure 6a). From this point on, all of these are actually commands (not a status read). (G) is the last command sent to HF-2 (Zynq 7020), while (H) shows, if present, the error reported by the clock board (DDS8m), otherwise “QUE\r\n” indicates no error, (I) HF-2 informs the operator that there is an internal error (serial port error between Zynq 7020 and the clock board), (J) the ADS1252 count number for 100 μV, (K) to initialize display filters and (L) to turn ON/OFF the display filters (M, N, O, P, and Q) are N/A (for debugging purposes).

#### 2.5.5. TCP/IP Packet

Upon startup, the RT software continuously monitors for a TCP/IP connection on port 6310; upon establishing a connection, it monitors the connection for command messages from the Host. A complete list of commands can be seen in Table 2. If a connection is lost or closed, the TCP/IP communication link goes back to monitoring for a new connection (Table 2). When requested via the TCP/IP command, the DMA FIFO is enabled, and data are read in blocks of 200 scans and transferred to the TCP/IP communications, from where payload packets of either 600 scans or 1400 scans are collected. The packet size is adjusted for performance, and the selection is determined by the sample rate that has been selected. Sample rates above 1000 samples per second utilize the larger packet size. A scan consists of the 8 channels of ADC data and the Sync state for a total of 9 channels. The complete format of the TCP/IP message can be seen in Table 3. 

#### 2.5.6. TCP/IP Message Format

There are two types of TCP/IP messages: command/response and data (Table 3). Specific configurations for messaging between Windows software and HF-2 are shown in Figure 6. The Windows software constantly listens for messages from HF-2. This packet structure allows flexible data transmission, dividing the data sets into multiple packets. The LabVIEW real-time data acquisition software efficiently manages data packet decoding and parsing.

#### 2.5.7. SNR Testing

We compared the new HF-2 system to our previous Biopac MP160 system, which is a commonly used acquisition and analysis system for life science research. We collected 30 min of 1 Hz 180 mVpp sinusoid information, which was generated by a function generator (AFG1062, Tektronix, Beaverton, OR, USA) and connected to both systems through a BNC cable while both systems were running with the same sampling frequency (Fs = 1 kS/s). Two spectrograms were computed in MATLAB (periodogram) with a window of 512 samples after removing the DC component and normalizing both waveforms to 1, given the different data formats in the two systems (V is used in Biopac, whereas µV is used for HF-2).

## 3. Results

### 3.1. PTFOS Safety

Figure 7 illustrates the temperature profile of a brain phantom in contact with a PTFOS grid during 30 min of fMRI with a 4.7 T magnetic field. The fMRI acquisition generated a power of 62.5 W, consistent with typical outputs for a full-body coil. The temperature beneath the two PTFOS electrodes (Electrodes 1 and 2) increased by 0.21 °C compared to the center of the phantom (core). This increase is within the 0.25 °C measurement error margin of the temperature sensors, suggesting negligible heating at both the electrodes and the deeper central areas of the phantom. Consequently, the risk of heat-induced tissue damage can be considered negligible.

### 3.2. In Vivo Studies

The in vivo rat brain imaging at 9.4 T did not exhibit appreciable signal loss adjacent to the PTFOS electrodes in the anatomical images when using the T1W, T2W, PD, FLASH, and UTE sequences (Figure 8). The signal loss on top of the brain in the FLASH image was due to the high sensitivity of the imaging sequence (FLASH) to the susceptibility mismatch induced by the surgical procedure, but not because of the PTFOS device. We also obtained good diffusion tensor fiber tracks in the PTFOS rat images while the connector was present. Compared to the rat with sham implantation, neither the sham Gelfilm nor the PTFOS grid introduced appreciable artifacts to the images at 9.4 T. To assess the impact of the PTFOS electrodes on imaging quality, we examined the SNR in the T2-weighted images. Figure 9 and Table 4 showed comparable SNRs among the sham and PTFOS implantations, in both the brain and the entire imaging volume. For fMRI, the minimal distortion observed in the EPI images in the PTFOS and sham rats apparently resulted from the tissue trauma introduced by the implantation procedure, not because of the PTFOS device. We were able to obtain reasonable fMRI activity readings in response to forepaw stimulation (2 Hz/1 mA) of the rat with the PTFOS implantation (Figure 10).

### 3.3. Shield Testing

The HF-2’s onboard processors, along with various digital signals, can degrade the MRI imaging quality if they are not meticulously shielded. To combat this, both the chassis and cabling of the HF-2 system have been enhanced with double-shielding to diminish the RF noise coupling significantly. Illustrated in Figure 11 is a clear demonstration of how the HF-2’s enhanced shielding positively impacts EPI (fMRI) images. The images (SW 200 kHz, TE 12.77 ms, TR 1 s, dimensions of 82 × 82 mm, FOV of 35 × 35 mm, a slice thickness of 1 mm, and a resolution of 0.43 × 0.43 mm across 25 slices) were captured under three identical conditions. This comparison vividly underscores the shielding’s efficacy. During experiments, when the ADC chassis’s cover was removed, a notable degradation in image quality was observed (Figure 11a), illustrating the critical role of shielding. These compromised images starkly contrasted with those obtained under conditions without concurrent electrophysiological recording, highlighting the interference caused by unshielded electronic components. Conversely, in sessions where the shielding was reinstated correctly, the image’s noise levels reverted to baseline, affirming the effectiveness of the HF-2’s shielding (Figure 11b). For these tests, the animal was a male rat weighing 306 g, which had been anesthetized using an alpha-chloralose bolus (80 mg/kg; IP) to facilitate mechanical ventilation alongside pancuronium for muscle paralysis (2 mg/kg; IV), ensuring consistency and reliability in the imaging results.

### 3.4. Synchronization Testing of HF-2 

The HF-2 leverages the capabilities of the Novatech synthesizer card, specifically the DD8sm model, as depicted in Figure 3 and Figure 4c. This integration synchronizes the internal clock of the HF-2 with the master clock of the MRI scanner, operating at a frequency of 10 MHz. The front panel of the HF-2 is equipped with a switch, affording the user the flexibility to transition seamlessly between the internal and external clock sources. The profound impact of this synchronization is distinctly illustrated in Figure 12. Observable benefits emerge when the HF-2 and the MRI scanner clocks are harmoniously synchronized. The top panel of Figure 12 demonstrates a significant reduction in both EPI noise and variance, thereby optimizing the performance of the HF-2 system. 

In contrast, the bottom panel of Figure 12 illustrates the conditions when no synchronization is applied, highlighting the substantial increase in noise and variance.

This synchronization mechanism enhances the signal quality of the HF-2 during fMRI and underscores its adaptability in terms of producing high-quality electrophysiological signals. The ability to minimize EPI noise and variance through synchronization contributes to the overall efficiency and SNR of the HF-2, making it a robust solution for applications demanding superior signal fidelity in conjunction with MRI procedures.

### 3.5. SNR Testing of HF-2 

We conducted a comparative analysis between the new HF-2 system and the Biopac MP160 system. The resulting spectrum revealed that the HF-2 system boasts a significantly higher signal-to-noise ratio (SNR) compared to the Biopac system (Figure 13). This higher SNR is evidenced by a cleaner background and fewer spurious peaks in the HF-2’s spectrum. The superior performance of the HF-2 system indicates its enhanced capability to provide clearer and more accurate data, which is crucial for precise and reliable measurements in ultra-high-field MRI.

## 4. Discussion

### 4.1. PTFOS

Conventional grids do not optimally contact the convoluted surface of the brain, due to their rigidity and large thickness/size, reducing the efficacy of recording and stimulation and possibly increasing the risk of tissue injury. Conversely, PTFOS has a 15-µm thin and flexible design. Different staining methods showed minimal injury in the brain tissue underlying PTFOS, suggesting that PTFOS may lead to fewer surgical complications than conventional grids. Additionally, the PTF printing technology allows for re-designing PTFOS grids specifically for a particular animal model of cortical anatomy, to improve cortical recording and stimulation [9,10,11]. These features also make PTFOS an attractive choice for brain–machine interfaces [12].

The recording of high-fidelity ECoG-fMRI has remained limited in small animal research, and very few publications justify that limitation. In the absence of an adequate tool to reliably acquire ECoG-fMRI data, researchers in the field cannot perform studies as they do for human EEG-fMRI data. The existing MRI-compatible headstages on the market for small animals are inadequate for acquiring high-fidelity EEG or ECoG-fMRI data. Using a standard fMRI sequence, the heating results demonstrate that the extremely low temperature does not cause any cortical damage or produce any unwanted functional stimulation. Future research could explore the use of Gelfilm with varying thicknesses and durometers to assess their potential impact on image quality. 

This research extends the groundwork laid in our earlier study published in *Radiology* [2], where we demonstrated that the PTFOS technology operates without artifacts, even at higher MRI field strengths. Specifically, our previous investigations at 7 T have now been successfully improved by showing applications at 9.4 T, confirming the technology’s robustness and reliability in even more demanding environments. Additionally, our prior research was limited to post-mortem imaging applications [2]. 

In a significant expansion of our previous research [2], the current study comprehensively explores various MRI imaging techniques applied to a live rodent. We utilized an array of techniques, including T1-weighted, proton density, T2-weighted, fast low-angle shot (FLASH), ultrashort echo time (UTE), diffusion tensor imaging (DTI), and functional MRI (fMRI) on a live rat with an implanted PTFOS device. This approach demonstrates the PTFOS technology’s versatility in a dynamic, living organism and showcases its potential to facilitate a wide range of high-resolution diagnostic imaging tasks. This advancement opens new avenues for real-time, in vivo research and clinical applications, potentially transforming how biological processes are monitored and how diseases are diagnosed at high magnetic-field strengths.

The safety profile and long-term biocompatibility of the materials used in this study have been previously reported in Radiology [2] (Figure 14). Briefly, our review of five different histologic stains demonstrated significant neighboring brain tissue damage from the use of conventional grids, while the gelatin film and PTFOS grids caused minimal or no damage. Specifically, we employed the following stains: hematoxylin-eosin (for assessing gross tissue disruption) Fluoro-Jade B (for detecting neuronal death and degeneration), NeuN (for measuring the density of living neuron nuclei), ionized calcium-binding adapter molecule 1 (for gauging microglia density and inflammation in the cortex), and silver staining (for evaluating the integrity of cortical nerve fibers). According to an independent neuropathologist’s qualitative assessment (who was not an author), all staining methods indicated more tissue injury due to conventional grids compared to the gelatin film and PTFOS grids. The tissue in contact with the PTFOS grid and gelatin film appeared similar across all five stains, highlighting their superior biocompatibility. The tensile mechanical properties of PTFOS were evaluated in a Radiology paper [2] by three neurosurgeons. They consistently rated the tensile strength, flexibility, and ease of handling of PTFOS grids as 2, 1, and 1 (top grades), respectively. The interclass correlation among the neurosurgeons’ ratings was perfect, with a value of 1. The tensile strength of the PTFOS and conventional grids showed no significant difference (χ^2^ test, *p* > 0.05). However, the PTFOS grids were rated significantly better in terms of flexibility and ease of handling compared to conventional grids (χ^2^ test, *p* < 0.05). 

Additionally, chronically implanted PTFOS disks in rodents were examined for microstructural stability using a scanning electron microscope, which revealed no changes before or after chronic implantation (Figure 15 , previously reported in Radiology [2]). The same study demonstrated that the electrical stability of PTFOS remained unchanged after long-term submersion in a saline bath.

Our previous study showed that PTFOS induced no appreciable artifacts on CT and MR images when used over a human cadaveric head specimen [2]. For comparison, the head specimen was also imaged without a grid and with conventional grids. To quantify the image quality, we enlisted board-certified neuroradiologists with decades of experience. They evaluated the images and rated the quality of images showing brain tissue overlaid with PTFOS grids significantly higher than those with conventional ECoG grids (two-tailed *t*-test, *p* < 0.05). This study highlights the superior imaging compatibility of PTFOS grids, which do not compromise image quality or introduce artifacts. This is particularly important for clinical and research settings where accurate imaging is critical. The neuroradiologists’ ratings underscore the potential of PTFOS grids to enhance diagnostic precision and effectiveness in neuroimaging applications.

### 4.2. HF-2

The HF-2 system is a versatile open-source software designed for the acquisition and real-time processing of electrophysiology during fMRI. Its open-source nature allows end users to extensively customize the software to match the specific needs of their applications. Specifically engineered for ultra-high frequency (UHF) MRI applications, the HF-2 offers a robust data acquisition hardware template that fulfills the rigorous and costly standards of MRI safety, compatibility, and electromagnetic compliance.

The software’s integrated subsystems are fully customizable across various levels of complexity, from low-level processors and FPGA firmware to sophisticated high-level data acquisition and analysis software. This adaptability enables end users to develop tailored real-time acquisition and processing applications quickly. There is an increasing demand in both neuroscience and clinical medicine for such innovative real-time solutions, particularly for use in epilepsy monitoring, neurofeedback, sleep studies, and electrophysiological drug trials.

Hardware for simultaneous EEG and fMRI acquisition has been developed independently across various laboratories [13,14,15,16]. Additionally, hardware for electrophysiological recordings during fMRI in animals has been created [17]. However, none of these systems offer a comprehensive open-source solution that integrates MRI-compatible hardware with customizable, real-time software. Currently, there are two main commercially available systems designed for EEG or ECoG data acquisition during fMRI scans: the MR-compatible headstage by Tucker-Davis Technologies (TDT), which has been tested up to 7 T [18], and the MagRes headstage by Blackrock Microsystems. At Massachusetts General Hospital (MGH), we utilize the TDT system. The TDT headstage functions as a pre-amplifier, with a secondary amplifier located externally in the control room. While effective, this setup is susceptible to noise interference due to the pre-amplification of analog signals rather than digitizing at or near the source level—a feature that our HF-2 system incorporates, particularly when miniaturized into a headstage. A critical limitation of the TDT system stems from its reliance on the Intan technology chip, which lacks internal clocks or timers. [19]. The timing of Intan chips depends on the microcontroller or FPGA that issues SPI commands, with the fastest digitization rate at approximately 1.05 MSamples/second through a 16-bit SPI command, sent every 950 nanoseconds. This rate is insufficient for achieving synchronization with the MRI scanner’s 10 MHz clock, making it challenging to effectively remove gradient artifacts from the data recorded during fMRI acquisition. Accurate hardware synchronization between the electrophysiological recording and the MRI acquisition clocks is paramount for optimizing gradient artifact removal. The MagRes system by Blackrock Microsystems also employs the Intan chip, thereby inheriting similar synchronization challenges. While MagRes offers 16 channels, our proposed system boasts a 32-channel capacity, providing a higher channel count for more comprehensive data acquisition. It is worth noting that most other European ECoG headstages also use Intan chips and face difficulties in meeting the stringent requirements for MRI recording. Table 5 shows a summarized comparison of the features and limitations of these systems in relation to our proposed HF-2 system, highlighting the advancements and increased capabilities that we aim to bring to EEG/ECoG acquisition during fMRI scans [20]. 

The OpenEEG Project (http://openeeg.sourceforge.net/doc/, accessed on 1 August 2024) presents an open-source EEG hardware system aimed at facilitating low-cost EEG acquisition; although it lacks MRI-specific design features, detailed specifications are given for automated hardware assembly and the level of software integration provided by the HF-2 system. In contrast, the HF-2 system, developed within the user-friendly LabVIEW graphical programming environment, supports the rapid development of real-time signal processing, data acquisition, and display modules that are fully tailored to specific user needs. Furthermore, standalone systems and software for real-time EEG/fMRI processing have been noted in the literature [16,22]. The HF-2 system’s software stands out for its capacity to support extensive customization and its ease of use, which are crucial for efficient and effective experimental setups. Real-time signal processing has essential applications, such as verifying the adequacy of data recorded during experiments. For instance, in simultaneous ERP/fMRI studies, real-time ERP averaging can assess ERP signal quality to ensure that sufficient ERP epochs are captured. This real-time analysis helps identify suboptimal subject arousal, excessive movement, or high electrode impedance—factors that can degrade signal quality and that are often hard to detect during experiments. Immediate problem identification and resolution can save considerable time and effort, enhancing the experimental outcomes. For studies focusing on sleep, anesthesia, or resting-state EEG/fMRI, which often analyze EEG spectral content, real-time frequency spectrum analysis is invaluable for monitoring the subject’s state. In this paper, we demonstrate the utility of both real-time ERP averaging and frequency-domain analyses, including real-time 40-Hz ASSR calculations, illustrating the robust capabilities of the HF-2 system in supporting advanced neuroscience research.

The HF-2 instrument represents a significant leap forward from the HF-1 [21]. While the HF1 relied on an 8-bit microcontroller and assembly language, the HF-2 leverages field-programmable gate array (FPGA) technology from National Instruments. This allows for more flexible hardware customization and higher performance. Additionally, the HF-2 utilizes software developed through graphical programming, offering a more user-friendly development experience compared to assembly language. The HF-2 also runs a real-time operating system, enabling more complex operations and offering improved multitasking capabilities.

### 4.3. Limitations

In this study, we did not include in vivo ECoG/fMRI data since the primary focus was on imaging, and a previous study already reported that oscillatory local field potentials, captured with cortical recordings and cortical stimulations in the motor cortex, elicited muscle contractions, with both captured in rodents [2]. Additionally, we have yet to present data on long-term stability, despite having previously published a study on a similar implant in our *Radiology* paper [2]. The emphasis here is on the imaging aspects of HF-2 rather than on the integration of ECoG/fMRI data or the assessment of long-term implant performance. Furthermore, the current open-source distribution contains code exclusively written in LabVIEW, covering programming for all three environments: Windows, the embedded processor, and the FPGA. While LabVIEW is robust, it is less widely used than other programming languages such as MATLAB. Despite this, it is possible to control the CompactRIO using MATLAB, though the methods to achieve this are more complex. However, data communication between MATLAB and the HF-2 can still be readily achieved using TCP, allowing for MATLAB’s advanced data analysis capabilities to be integrated with the HF-2 system.

### 4.4. Future Work

We are currently focused on developing a plugin in C++ to seamlessly integrate continuous HF-2 data into the Open Ephys’ existing signal chain. The Open Ephys GUI is a pioneering and widely used application for multichannel electrophysiology that leverages a plugin-based workflow system. This integration will allow users to utilize current plugins for various functionalities, including displaying the data, conducting real-time analysis, and saving it in multiple formats, such as the Neurodata Without Borders (NWB) standard. This integration process is designed for efficiency, ensuring full functionality without requiring additional development efforts. Open Ephys will provide seamless compatibility with the Data Portal “the DANDI” Archive, fulfilling NIH Data Management and Sharing Policy (DMSP) requirements. Furthermore, Open Ephys includes an open-source simulation management platform called StimJim, which will be compatible with HF-2. By developing this plugin, we aim to enhance the utility and versatility of HF-2 data, enabling researchers to take full advantage of the Open Ephys ecosystem for advanced electrophysiological studies. Finally, we plan to conduct future studies to evaluate the performance of PTFOS during simultaneous ECoG and fMRI acquisition with ultra-high-field MRI (UHF-MRI). Additionally, we aim to perform benchmark testing using electrical impedance spectroscopy and cyclic voltammetry. These tests will help us to thoroughly characterize PTFOS and compare its performance against commercial electrodes.

## 5. Conclusions

PTFOS grids are an attractive alternative for preclinical studies compared to conventional grids, due to their mechanical properties, MR heating profile, and MR imaging artifact issues. Preclinical small animal brain imaging makes valuable contributions to improving our understanding of human diseases such as epilepsy, sleep disorders, and traumatic brain injury, as well as for safety pharmacology studies to assess potential side effects during drug development, neuro-behavioral assessment, and spatial learning and understanding of memory processes. Multimodal imaging approaches combine fMRI with iEEG or ECoG. The HF-2 system introduced here is a comprehensive open-source hardware and software solution designed for electrophysiological recording during MRI procedures. This versatile system offers essential components for developing custom MRI-compatible data acquisition hardware. Additionally, it supports the rapid creation of tailored data acquisition and real-time signal processing tools within the LabVIEW environment, facilitating advanced research and development. The combination of fMRI with ECoG using PTFOS and the HF-2 system provides an unparalleled view of the global network activity across cortical areas, revealing information at high temporal and spatial scales about functional connectivity and the intrinsic oscillatory properties, which information is of vital importance in many branches of neuroscience and medicine. The advantage of combining EEG or ECoG with fMRI lies in its crucial role in preclinical research, providing essential insights into basic neuroscience and enhancing our understanding of neurological diseases. 

## Figures and Tables

**Figure 1 sensors-24-05214-f001:**
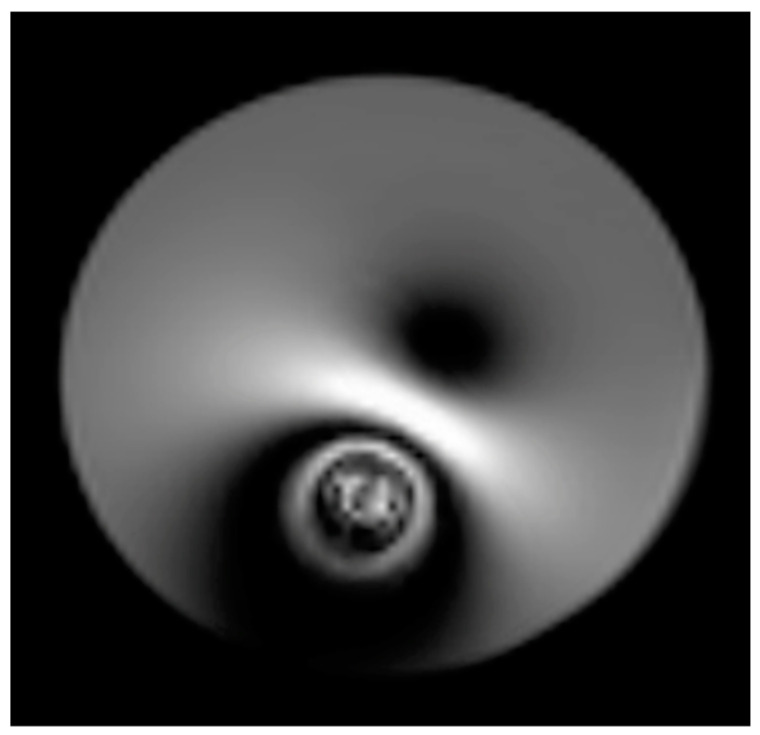
Susceptibility artifact induced by a Pt wire (common implant wires) in a phantom on an ultra-high magnetic field MRI (9.4 T).

**Figure 2 sensors-24-05214-f002:**
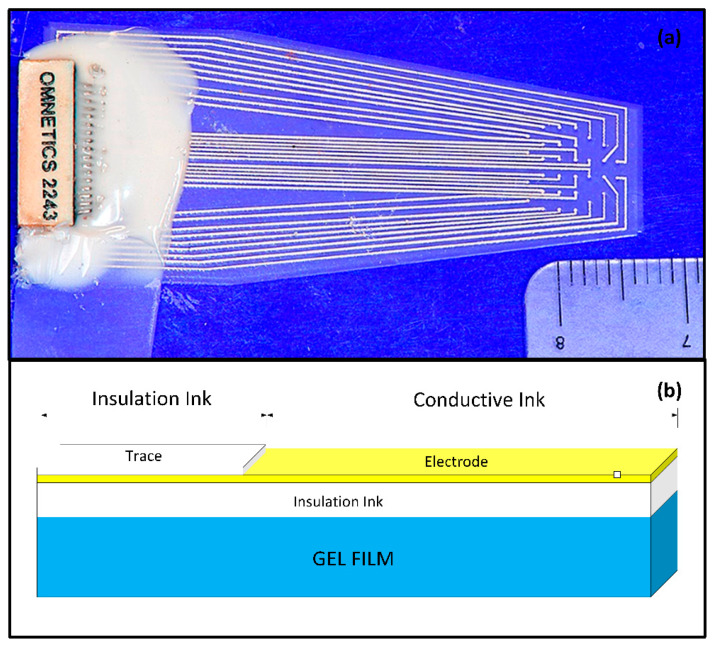
The PTFOS. (**a**) Image of the 32 electrodes and the connector. (**b**) The PTFOS layout is based on an absorbable gelatin film made from denatured collagen (Gelfilm by Pharmacia and Upjohn Co, Division of Pfizer Inc., New York City, NJ, USA).

**Figure 3 sensors-24-05214-f003:**
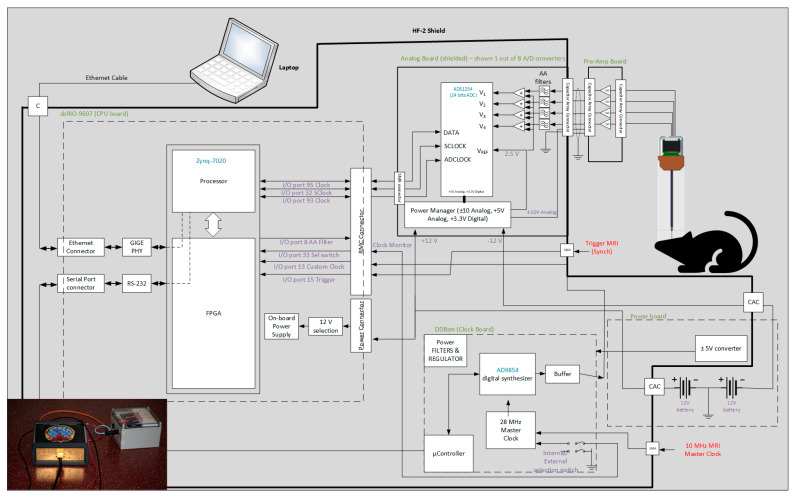
HF-2 system architectural diagram [7]. HF-2 is composed of four boards inside a shielded enclosure, with an external preamplification board. Below is an image of the HF-2 system and the battery.

**Figure 4 sensors-24-05214-f004:**
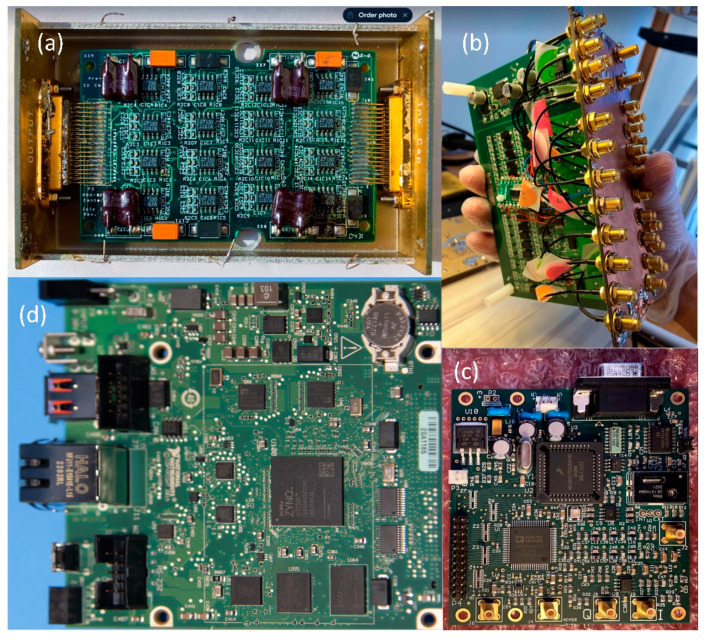
HF-2 hardware [7]: (**a**) preamplification board, (**b**) analog board, (**c**) clock board, and (**d**) CPU board.

**Figure 5 sensors-24-05214-f005:**
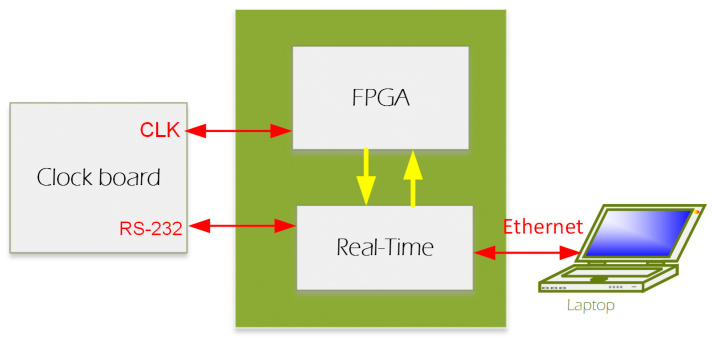
Real-Time software.

**Figure 6 sensors-24-05214-f006:**
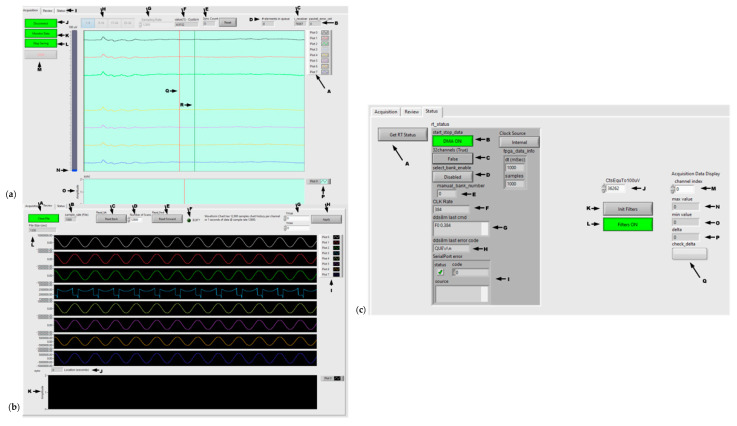
Windows LabVIEW: (**a**) Host interface showing screenshots obtained from the user interface in a rat experiment. The PTFOS was implanted into the right sensory S1 cortex, and stimulation electrodes were placed in the right and left forelimbs and hindlimbs. (**b**) The review panel will look at the existing recordings (i.e., sinusoid). (**c**) Status bar.

**Figure 7 sensors-24-05214-f007:**
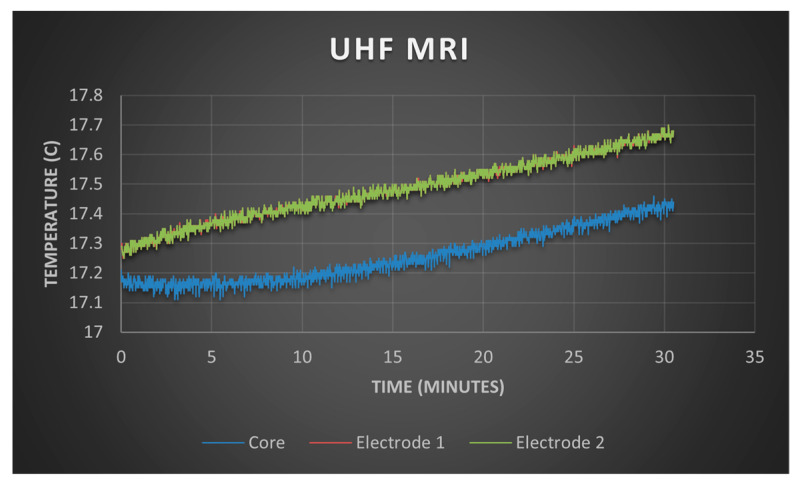
The temperature test of the PTFOS.

**Figure 8 sensors-24-05214-f008:**
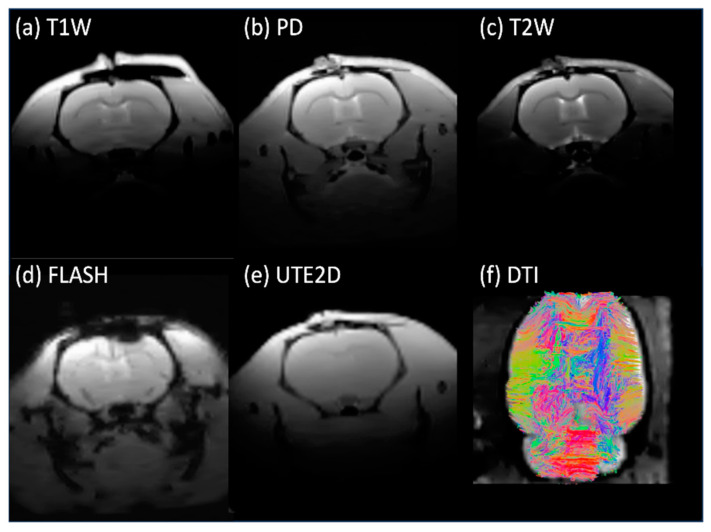
In vivo images of the rat with the PTFOS and connector implanted at 9.4 T, showing only minimal artifacts near the electrodes.

**Figure 9 sensors-24-05214-f009:**
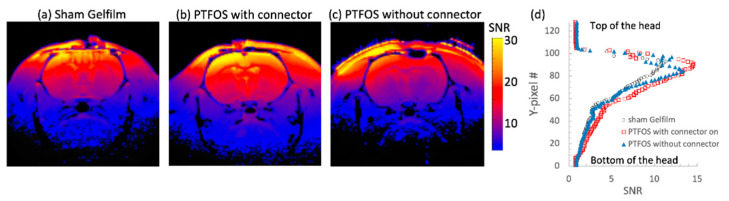
SNR comparison of T2-weighted images of rats with (**a**) sham Gelfilm, (**b**) PTFOS and a connector, and (**c**) PTFOS without a connector. (**d**) The SNR projection, from the top to the bottom of the head. The SNR on the top portion of the brain was relatively smooth and homogeneous with the PTFOS with a connector, compared to the ones without the connector or with sham Gelfilm.

**Figure 10 sensors-24-05214-f010:**
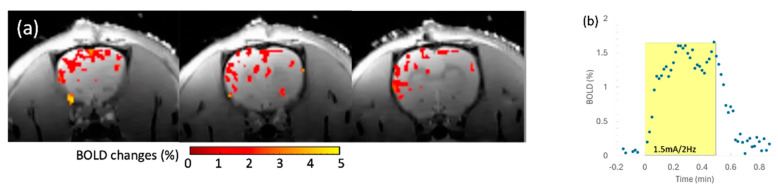
The 9.4 T fMRI with electrical forepaw stimulation (1.5 mA/2 Hz) on a rat with PTFOS implanted. (**a**) The colormap representing the brain area, with statistically significant BOLD responses to the forepaw stimulation. (**b**) The BOLD time course in response to the electrical forepaw stimulation (yellow block).

**Figure 11 sensors-24-05214-f011:**
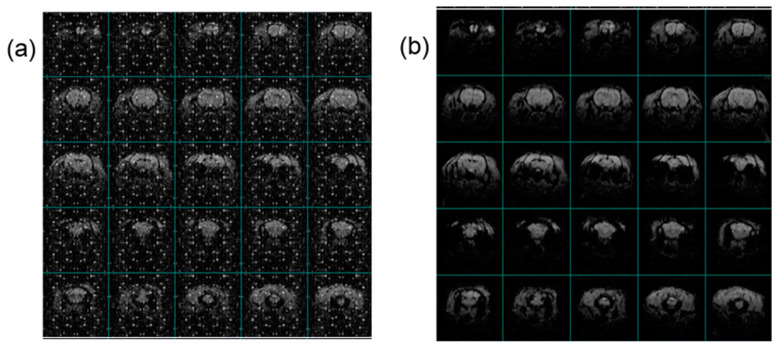
Shielding of the HF-2, showing the effect on EPI (fMRI) images at 4.7 T: (**a**) System ON and Shielding OFF, and (**b**) System and Shielding ON.

**Figure 12 sensors-24-05214-f012:**
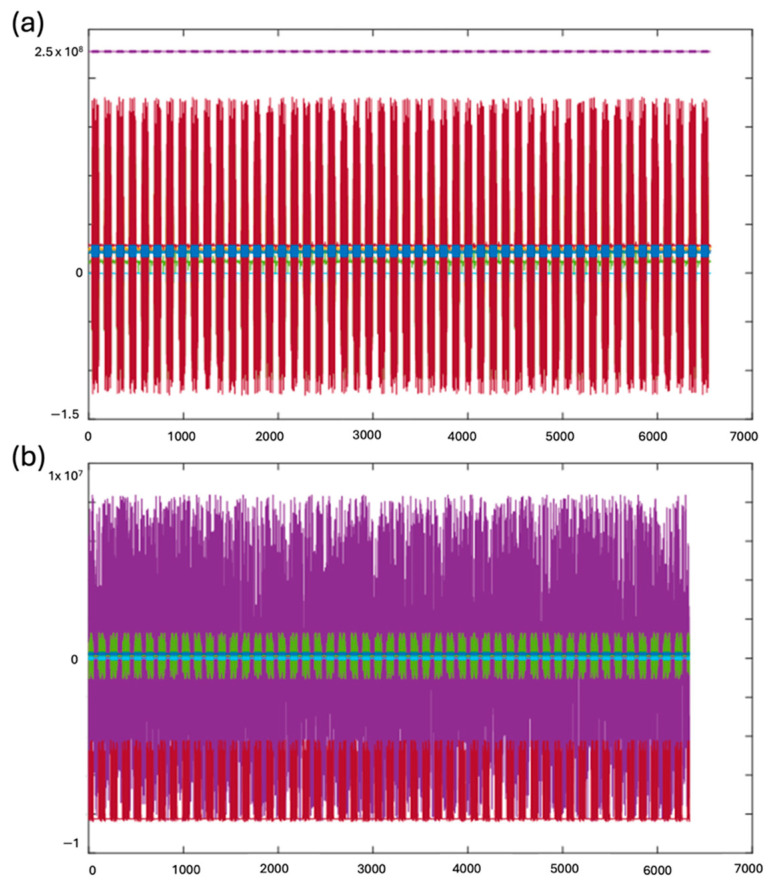
Synchronized vs. out-of-synchronization recordings of HF-2 with a phantom at 4.7 T (Brucker), using EPI. (**a**) Eight channels of HF-2 recordings, with the system clock synchronized to the 4.7 T MRI scanner. (**b**) Recordings without synchronization to the MRI’s master clock. These raw data illustrate that without synchronization, the recording is affected by more EPI noise and more variance.

**Figure 13 sensors-24-05214-f013:**
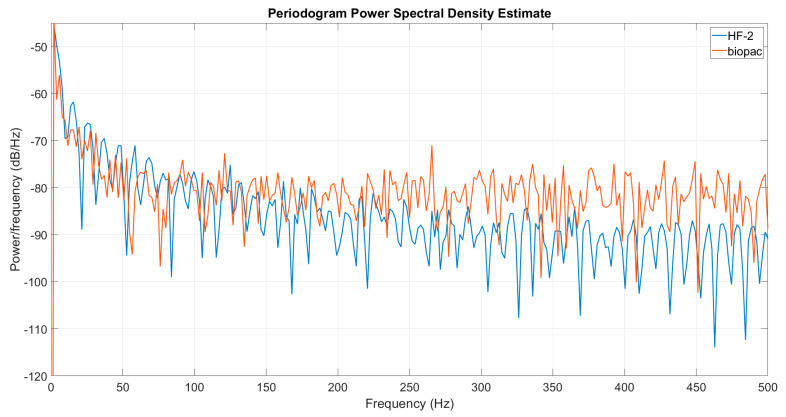
The spectrogram of a sinusoid acquired with HF2 (blue) and Biopac (orange). The Biopac system has a higher noise floor and exhibits more noise peaks.

**Figure 14 sensors-24-05214-f014:**
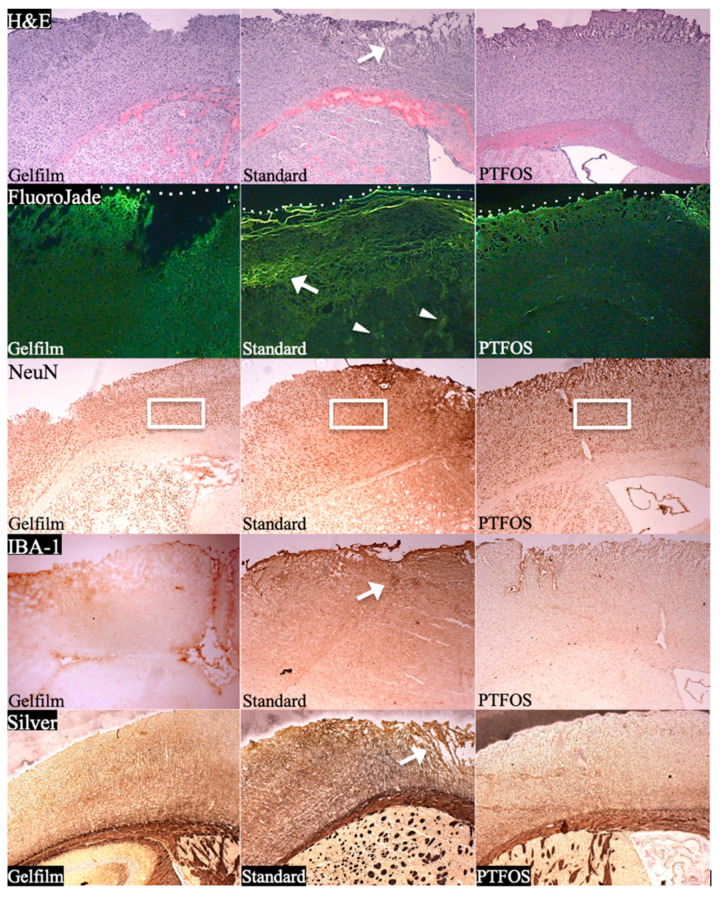
Tissue reactivity of PTFOS compared to Gelfilm^®^ and a conventional grid. H&E panel shows greater tissue disruption (arrow) from the standard grid as compared to Gelfilm^®^ or PTFOS. Fluoro-Jade panel shows bright lines of injured cells in the cortex (arrow) in contact with the standard grid as well as patches of injured cells in subcortical tissues (arrowheads). White dots indicate the interface between the implant and the cortical surface. Minimal injury is seen in the images of the tissues in contact with Gelfilm^®^ and PTFOS. NeuN panel shows lower density of neuronal nuclei in the cortex in contact with the conventional grid as compared to Gelfilm^®^ and PTFOS (compare the density of nuclei inside the frames). IBA-1 panel shows higher density of microglia in the cortex in contact with the conventional grid as compared to Gelfilm^®^ and PTFOS (the arrow shows an area of microglia accumulation). Silver panel shows more disruption of cortical nerve fibers (arrow) from the conventional grid as compared to Gelfilm^®^ or PTFOS. The images of the tissue neighboring PTFOS and Gelfilm^®^ appear the same across all staining methods. This figure was previously reported in Radiology [2].

**Figure 15 sensors-24-05214-f015:**
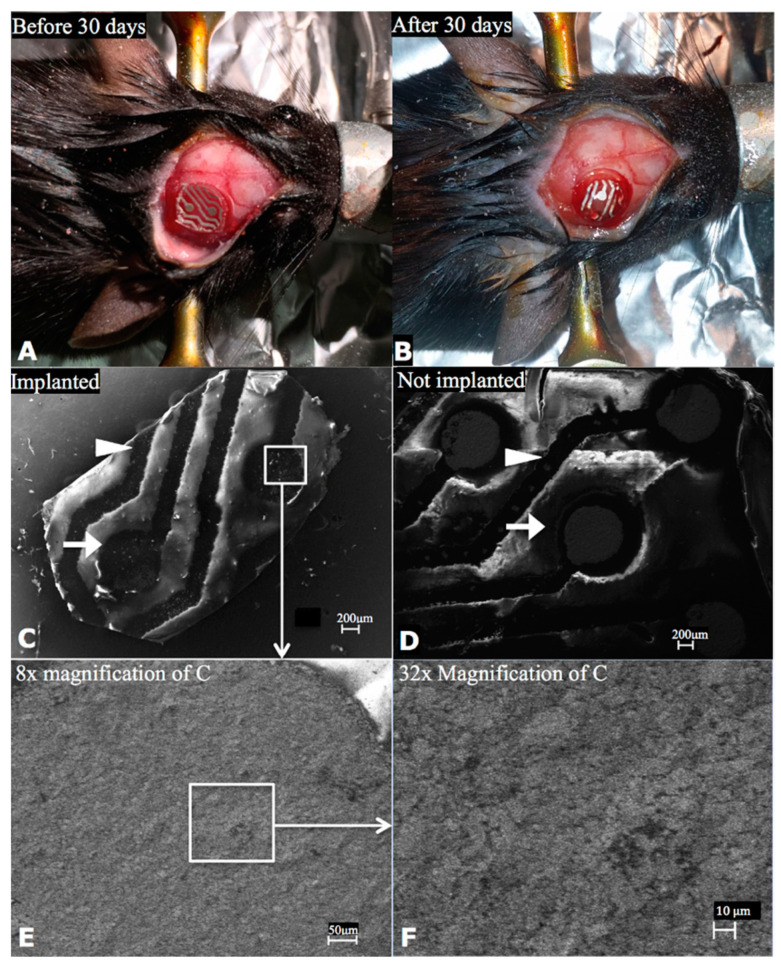
Microstructural stability of PTFOS implants. We implanted 5mm disks of PTFOS over the cortex of two mice (**A**). After 30 days, we harvested the PTFOS implant (**B**) and imaged it using a scanning electron microscope (**C**). For comparison, we also imaged a PTFOS disk that was not implanted (**D**). We used high-magnification levels, enabling us to see the PTFOS microstructure with the scale of 10 μm (**E**,**F**), to search selected areas of electrodes (arrows) and conductive lines (arrowheads) for the presence of breaks/cracks. We did not find any breaks/cracks in the electrodes/conductive lines of the PTFOS implants that were and were not implanted. This figure was previously reported in Radiology [2].

**Table 1 sensors-24-05214-t001:** Parameter settings for each sequence used in the 9.4 T imaging session. FOV: field of view. TR: repetition time. TE: echo time. T1W-TSE average = 2. UTE flip angle = 15°. DTI delta = 3 ms, DELTA = 10.6 ms, B = 700 s/mm^2^, diffusion directions = 36. fMRI slice thickness = 1 mm.

	FOV (mm × mm)	Dimension	Resolution (mm × mm)	TR (ms)	TE (ms)	Slices
Proton Density (PD)	35 × 35	128 × 128	0.273 × 0.273	2192	8/52	22
T2-Weighted (T2W)	35 × 35	128 × 128	0.273 × 0.273	2192	8/52	22
T1-Weighted TSE	35 × 35	128 × 128	0.273 × 0.273	1500	7.5	22
FLASH	35 × 35	128 × 128	0.273 × 0.273	326	7.5	22
UTE	35 × 35	128 × 128	0.273 × 0.273	30	0.457	22
DTI	35 × 35	96 × 96	0.36 × 0.36	1416	23.9	44
fMRI	35 × 35	92 × 92	0.38 × 0.38	500	12.11	18

**Table 2 sensors-24-05214-t002:** The HF-2 handshake command formats.

Command	Description	Reply
Set Frequency	Sets the sampling rate for the data acquisition	Ack/Nak
Select Bank Num	Selects the bank of 8 channels to acquire	Ack/Nak
Reset Sync Cnt	Resets the sync pulse counter on the FPGA to 0	Ack/Nak
Start/Stop	Start/Stop the TCP/IP data packet transfer from the RT to the Host	Ack/Nak
RT Status	Returns the status of the DMA FIFO (on/off), clock source (internal/external), CLK rate, serial port status	Status (see Figure 6c)

**Table 3 sensors-24-05214-t003:** Data messages.

**Bytes Indices**	**Description (message type 100 or command/response)**
**0–3**	Fixed value of ‘100’ that identifies the message type
**4–7**	Length of payload
**8+**	Payload; command settings/data
**Byte Indices**	**Description (Message type 101 or a data message)**
**0–3**	Fixed value of ‘101’ that identifies the message type
**4–7**	Length of payload
**8–11**	Sequence Number
**12–15**	Sync Counter Value
**16+**	Payload, 600 or 1400 scans of long integer (4 bytes or 32 bits to send 24 bits) sending ADC 8 channels plus 1 trigger channel in a round-robin format (0, 1, 2, 3, 4, 5, 6, 7, 8, 0, 1, 2, 3, 4, 5, 6, 7, 8, etc.)

**Table 4 sensors-24-05214-t004:** SNR from the brain area shown in Figure 9 and the whole imaging volume.

	Brain	Whole Imaging Volume
Sham (Gelfilm)	15.25 ± 5.63	11.77 ± 5.28
PTFOS w connector	18.66 ± 6.42	15.08 ± 5.54
PTFOS w/o connector	13.46 ± 5.28	12.19 ± 4.89

**Table 5 sensors-24-05214-t005:** Comparison with commercial preclinical electrophysiology systems.

Manufacture/Product	# of Channels	Amplifier and A/D Chip	# of Bits (Resolution)	Synchronization with MRI Clock (10 MHz)	MRI Compatibility
Bmseed/ECoG Electrode Array	65	Intan	16	No	No
Cambridgeneurotech/Mini-Amp-64	64	Intan	16	No	No
RippleNeuro/Nano-2	32	Intan	16	No	No
atlasneuro/AtlasNeuro	64	Intan	16	No	No
NeuroNexus/MRI compatible	64	Intan	16	No	No, only the electrodes
plexon/OmniPlex System	64	Intan	16	No	No
microprobes/Micro-electrode	32	Intan	16	No	No
Blackrockmicro/MagRes headstage	16	Intan	16	No	Yes
TDT/Tucker-Davis Technologies	64	Intan	22	No	Yes (up to 14 T)
MGH HF-1 [21]	32	ADS1254	24	Yes	Yes (up to 7 T)
eMRiSystems/HF-2 (benchmark)	32	ADS1298	24	Yes	Yes (up to 9.4 T)

## Data Availability

The complete software release packet of HF-2 can be found at: https://github.com/eMRISystems2024/eMRISystems_HF-2 (accessed on 5 August 2024).

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
