# Peer review of "A Polymer Thick Film on an Organic Substrate Grid Electrode and an Open-Source Recording System for UHF MRI: An Imaging Study"

_sensors, 2024, doi:10.3390/s24165214_

Round 1

Reviewer 1 Report

Comments and Suggestions for Authors

The manuscript presents a new grid electrode design aimed at UHF MRI procedures, based on a polymer thick-film deposited on an organic substrate. The relevance is clear and the English language is fine. There are several metrological issues, as follows:

- throughout the manuscript, units must be separated from the values (e.g. "9.4 T" instead of "9.4T". "500 um" instead of "500um", etc);

- when written in full, the units are in lower case (e.g. "4 teslas" instead of "4Tesla");

- there should be no hyphen in 12-V;

- there is no unit "sec";

- the frequency unit is kHz, not KHz;

- the sentence "accurate to 0.25°C." in line 189 makes no metrological sense, as accuracy is a qualitative term;

Other issues include:

- the entire content between lines 151 and 172 sounds like marketing material from NI, and provides too many details not needed for the manuscript;

- the first 3 columns (FOV, dimensions and resolution) of Table are very confusing;

- Table 2 is badly positioned within the manuscript;

- the entire section 2.5.7 about TCP/IP is too detailed for the scope of the manuscript and of the journal;

Author Response

Critique-1: Throughout the manuscript, units must be separated from the values (e.g., "9.4 T" instead of "9.4T". "500 um" instead of "500um", etc.). When written in full, the units are in lower case (e.g. "4 teslas" instead of "4Tesla"). There should be no hyphen in 12-V. There is no unit "sec". The frequency unit is kHz, not KHz.

Response-1: We thank the reviewer for their corrections. We separated all the units from the values, changed the units to lowercase when written in full, and removed hyphens from the manuscript as suggested. The “sec” units have been corrected. The KHz has been changed into kHz.

Critique-2: The sentence "accurate to 0.25°C." in line 189 makes no metrological sense, as accuracy is a qualitative term.

Response-2: We thank the reviewer for pointing out this issue since we realized, looking back at our probes, that the calibrated accuracy was actually better than what was originally reported in the paper. The text now writes:

Temperatures at the core and superficial areas of the phantom near the PTFOS grid electrodes were measured using eight optical sensors (OSENSA Innovations, Coquitlam, BC, Canada), which had a calibrated accuracy of 0.1 °C (PRB-400 Fiber Optic Temperature Probe).

We respectfully disagree with the reviewer about the definition of accuracy, as the Oxford Dictionary defines accuracy as a quantity and not a qualitative attribute: “Accuracy is how close a given set of measurements are to their true value. “ Furthermore, the Company assures this accuracy value of 0.1 °C for temperature ranges between 10°C and 60°C (see chrome-extension://oemmndcbldboiebfnladdacbdfmadadm/https://www.osensa.com/pdf/DATASHEET-DMK-0064A-1%20PRB-400%20PROBE.pdf).

Critique-3: The entire content between lines 151 and 172 sounds like marketing material from NI, and provides too many details not needed for the manuscript.

Response-3: We thank and agree with the Reviewer. The text now reads:

“(iii) CPU Board: The CPU board (Figures 3 and 4d) is engineered to bridge the Analog and Clock boards with a computer board (CompactRIO , National Instrument) to enhance integration and performance. The CPU board has an embedded FPGA controller and a real-time processor with NI Linux Real-Time and integrated Gigabit Ethernet, which is used to communicate and transfer data to the Windows computer/laptop. Auxiliary digital input signals, like triggers, are acquired by the CPU board and accessible from the front panel, synchronizing data acquisition with the MRI scanner's TTL pulse and signaling the start of each fMRI volume acquisition. This signal synchronizes data acquisition with the MRI scanner's TR or trigger pulse, a TTL pulse that signals the start of each volume acquisition in fMRI. The HF-2 is synchronized with the MRI acquisition system through the analog Analog board synchronization, which was implemented as follows.”

Critique-4:  The first 3 columns (FOV, dimensions and resolution) of Table are very confusing.

Response-4: We revised the table to make it more readable.

Critique-5: Table 2 is badly positioned within the manuscript.

Response-5: We repositioned Table 2 within the manuscript.

Critique-6: The entire section 2.5.7 about TCP/IP is too detailed for the scope of the manuscript and of the journal.

Response-6: The entire paragraph 2.5.7 has been reduced as requested by the Reviewer:

There are two types of TCP/IP messages: command/response and data (Table 3). Specific configurations for messaging between Windows software and HF-2 are shown in Figure 6. The Windows software constantly listens for messages from HF-2. This packet structure allows flexible data transmission, dividing data sets into multiple packets. The LabVIEW real-time data acquisition software efficiently manages data packet decoding and parsing.”

Reviewer 2 Report

Comments and Suggestions for Authors

Strengths:

  • Reduced Heating Profile: The PTFOS grid exhibits minimal heating during MRI scans, addressing a critical safety concern for ECoG-fMRI studies. This reduces the risk of tissue damage in the brain region of interest, improving animal welfare and the reliability of experimental data.
  • High-Fidelity Imaging: The PTFOS grid minimizes artifacts in various MRI sequences, including anatomical (T1w, T2w) and functional (fMRI) imaging. This allows researchers to obtain clear and undistorted images of the brain, facilitating accurate analysis of both structural and functional properties.
  • Open-Source and Customizable Hardware/Software: The HF-2 system is open-source, providing transparency and flexibility for researchers. Users can tailor data acquisition parameters and develop real-time processing modules specific to their experimental needs. This level of customization is advantageous for researchers working on diverse neuroscience questions.
  • Enhanced Synchronization: The HF-2 system synchronizes with the MRI scanner clock. This synchronization significantly improves the removal of scanner noise from the ECoG recordings. Cleaner ECoG data leads to more accurate measurements of brain activity during fMRI experiments.
  • Broad Applicability: The combination of PTFOS and HF-2 offers a valuable tool for preclinical research on various brain functions and diseases. By simultaneously recording ECoG signals with high spatial resolution and fMRI signals with excellent blood oxygen level-dependent (BOLD) contrast, researchers can gain deeper insights into the relationship between neuronal activity and brain function across different brain regions.

Weaknesses:

  • Limited In Vivo Data: The current study primarily relies on phantom data for demonstrating the system's functionality. While the results are promising, more in vivo validation with different animal models is necessary. Particularly, the authors should showcase simultaneously recorded fMRI and ECoG data from live animals. This would provide a more complete picture of the system's effectiveness in capturing real-time neurophysiological activity during MRI scans.
  • Software Accessibility: While the open-source nature of the HF-2 system is a strength, the use of LabView as the programming environment might limit accessibility for some researchers. LabView, though user-friendly for certain applications, is not as widely used in neuroscience research compared to other programming languages like Python or MATLAB. Exploring alternative or complementary software development options could broaden the user base.
  • Incomplete Comparison with Existing Systems: The paper compares the HF-2 system to existing commercial ECoG systems but focuses primarily on channel count. A more comprehensive comparison highlighting advantages and disadvantages beyond channel count, such as noise reduction capabilities, data processing features, and overall system cost, would be beneficial for researchers considering different ECoG-fMRI solutions.

Additional Considerations:

  • Chronic Implantation Safety: The authors should address the safety profile of the PTFOS grid for chronic implantation. Long-term biocompatibility studies are crucial for ensuring the grid's safety and suitability for extended use in animal models.
  • Long-Term Stability: Assessing the long-term stability of the PTFOS grid material is essential. Understanding how the material properties might change over time will inform researchers on the appropriate duration for implantation studies.
  • Open-Source Platform Integration: Exploring integration of the HF-2 system with existing open-source electrophysiology platforms, such as OpenBCI or Ephus, could foster wider adoption within the neuroscience research community. This would leverage existing software ecosystems and potentially streamline data acquisition and analysis workflows.

By addressing these weaknesses and incorporating the additional considerations, the authors can strengthen their contribution to the field of simultaneous ECoG-fMRI research. The PTFOS grid technology and HF-2 data acquisition system hold promise for advancing our understanding of brain function in health and disease.

Comments on the Quality of English Language

acceptable. 

Author Response

Critique-1: Limited In Vivo Data: The current study primarily relies on phantom data for demonstrating the system's functionality. While the results are promising, more in vivo validation with different animal models is necessary. Particularly, the authors should showcase simultaneously recorded fMRI and ECoG data from live animals. This would provide a more complete picture of the system's effectiveness in capturing real-time neurophysiological activity during MRI scans.

Response-1: We thank the reviewer and agree that ECoG/fMRI data in vivo is missing, but this was designed primarily as an imaging study. To make this clear to the reader, we changed the title, and the new title reads: “A Polymer Thick Film on an Organic Substrate Grid Electrode and an Open-Source Recording System for UHF MRI: An Imaging Study.” To emphasize our point of view, we added a new section called “4.3 Limitations” in the Discussion that reads as follows:

“In this study, we did not include in vivo ECoG/fMRI data as the primary focus was on imaging, and a previous study reported oscillatory local field potentials captured with cortical recordings and cortical stimulations in the motor cortex elicited muscle contractions both in rodents [2]. Additionally, we have yet to present data on long-term stability despite having previously published a study on a similar implant in our Radiology paper [2]. The emphasis here is on the imaging aspects rather than the integration of ECoG/fMRI data or the assessment of long-term implant performance.”

We are working on another manuscript that will present these data along with a more detailed study of the electrodes. This study will include ECoG/fMRI data, spectrum analysis of the electrodes, and cyclic voltammetry data compared with commercially available electrodes.

Critique-2: Software Accessibility: While the open-source nature of the HF-2 system is a strength, the use of LabView as the programming environment might limit accessibility for some researchers. LabView, though user-friendly for certain applications, is not as widely used in neuroscience research compared to other programming languages like Python or MATLAB. Exploring alternative or complementary software development options could broaden the user base.

Response-2: Unfortunately, MATLAB supports the low-level programming of our CPU board, as it can currently only be programmed using LabVIEW. In other words, there is no supported way to control a cRIO from MATLAB. However, there are ways to communicate data to or from the HF-2 with MATLAB using TCP. We added the following text to the limitations:

Furthermore, the current open-source distribution contains code exclusively written in LabVIEW, covering programming for all three environments: Windows, the embedded processor, and the FPGA. While LabVIEW is robust, it is less widely used than other programming languages such as MATLAB. Despite this, it is possible to control the CompactRIO using MATLAB, though the methods are more complex. However, data communication between MATLAB and the HF-2 can still be readily achieved using TCP, allowing for MATLAB's advanced data analysis capabilities to be integrated with the HF-2 system.”

Critique-3: Incomplete Comparison with Existing Systems: The paper compares the HF-2 system to existing commercial ECoG systems but focuses primarily on channel count. A more comprehensive comparison highlighting advantages and disadvantages beyond channel count, such as noise reduction capabilities, data processing features, and overall system cost, would be beneficial for researchers considering different ECoG-fMRI solutions.

Response-3: Table 5 outlines the differences between HF-2 and other systems, pointing out the number of channels and, most importantly, the number of bits of resolution. Furthermore, we have added another paragraph (3.5) and a new Figure (Figure 12) that shows the SNR of HF-2 vs. HF-1.

The text now reads:

“2.5.8. SNR Testing.

We compared the new HF-2 system to our previous Biopac MP160 system, which is a commonly used acquisition & analysis system for life science research. We collected 30 minutes of 1 Hz 180mVpp sinusoid generated by a function generator (AFG1062, Tektronix, Beaverton OR) and connected to both systems through a BNC cable while both systems were running with the same sampling frequency (Fs= 1 kS/s). Two spectrograms were computed in MATLAB (periodogram) with a window of 512 samples after removing the DC component and normalizing both waveforms to 1 given the different data formats in the two systems (V in biopic whereas µV for HF-2).

3.5 SNR testing of HF-2

We conducted a comparative analysis between the new HF-2 system and the Biopac MP160 system. The resulting spectrum revealed that the HF-2 system boasts a significantly higher Signal-to-Noise Ratio (SNR) compared to the Biopac system (Fig. 13). This higher SNR is evidenced by a cleaner background and fewer spurious peaks in the HF-2's spectrum. The superior performance of the HF-2 system indicates its enhanced capability to provide clearer and more accurate data, which is crucial for precise and reliable measurements in ultra-high field MRI.”

Figure 13: The Spectrogram of a sinusoid acquired with HF2 (blue) and Biopac (orange). The biopac system has a higher noise floor and exhibits more noise peaks

Critique-4: Chronic Implantation Safety: The authors should address the safety profile of the PTFOS grid for chronic implantation. Long-term biocompatibility studies are crucial for ensuring the grid's safety and suitability for extended use in animal models.

Response-3: The safety profile and long-term biocompatibility studies of the same materials used in this paper have been previously reported in Radiology. The Discussion now includes the following statement:

“The safety profile and long-term biocompatibility of the materials used in this study have been previously reported in Radiology [2]. Briefly, our review of five different histologic stains demonstrated significant neighboring brain tissue damage from conventional grids, while the gelatin film and PTFOS grids caused minimal or no damage. Specifically, we employed the following stains: Hematoxylin-eosin (for assessing gross tissue disruption) Fluoro-Jade B (for detecting neuronal death and degeneration), NeuN (for measuring the density of living neuron nuclei), Ionized calcium-binding adapter molecule 1 (for gauging microglia density and inflammation in the cortex), Silver staining (for evaluating the integrity of cortical nerve fibers). According to an independent neuropathologist's qualitative assessment (who was not an author), all staining methods indicated more tissue injury with conventional grids compared to the gelatin film and PTFOS grids. The tissue in contact with the PTFOS grid and gelatin film appeared similar across all five stains, highlighting their superior biocompatibility.”

For the Reviewer’s information, we include slides presented in July 2024 at the IEEE EMBC 2024 in Orlando, Florida. These data were extracted from the Radiology [2] paper.

 Please see figures in the attached file.

Critique-5: Long-Term Stability: Assessing the long-term stability of the PTFOS grid material is essential. Understanding how the material properties might change over time will inform researchers on the appropriate duration for implantation studies.

Response-5: The long-term stability of the same materials used in this paper has been previously reported in Radiology. The Discussion now includes the following statement:

” The tensile mechanical properties of PTFOS were evaluated in a Radiology paper by three neurosurgeons. They consistently rated the tensile strength, flexibility, and ease of handling of PTFOS grids as 2, 1, and 1 (top grades), respectively. The interclass correlation among the neurosurgeons' ratings was perfect, with a value of 1. The tensile strength of PTFOS and conventional grids showed no significant difference (χ² test, P > 0.05). However, PTFOS grids were rated significantly better in terms of flexibility and ease of handling compared to conventional grids (χ² test, P < 0.05).

Additionally, chronically implanted PTFOS disks in rodents were examined for microstructural stability using a scanning electron microscope, which revealed no changes before and after chronic implantation. The same study demonstrated that the electrical stability of PTFOS remained unchanged after long-term submersion in a saline bath.”

For the Reviewer’s information, we include slides presented in July 2024 at the IEEE EMBC 2024 in Orlando, Florida. These data were extracted from the Radiology [2] paper.

Critique-6: Open-Source Platform Integration: Exploring integration of the HF-2 system with existing open-source electrophysiology platforms, such as OpenBCI or Ephus, could foster wider adoption within the neuroscience research community. This would leverage existing software ecosystems and potentially streamline data acquisition and analysis workflows.

Response-6: We thank the Reviewer. We made this point at the end of the discussion. However, we have now rewritten this paragraph for improved clarity:

We are focused on developing a plugin in C++ to seamlessly integrate continuous HF-2 data into the Open Ephys' existing signal chain. Open Ephys GUI is a pioneering and widely used application for multichannel electrophysiology that leverages a plugin-based workflow. This integration will allow users to utilize current plugins for various functionalities, including displaying the data, conducting real-time analysis, and saving it in multiple formats, such as the Neurodata Without Borders (NWB) standard. This integration process is designed for efficiency, ensuring full functionality without requiring additional development efforts. Open Ephys will provide seamless compatibility with the Data Portal "the DANDI" Archive, fulfilling NIH Data Management and Sharing Policy (DMSP) requirements. Furthermore, Open Ephys includes an open-source simulation management platform called StimJim, which will be compatible with HF-2. By developing this plugin, we aim to enhance the utility and versatility of HF-2 data, enabling researchers to take full advantage of the Open Ephys ecosystem for advanced electrophysiological studies.”

Reviewer 3 Report

Comments and Suggestions for Authors

1- For broader readability, it would be beneficial to include brief explanations of technical terms and concepts, such as inversion recovery (IR) sequence and steady state signal equation. This would make the manuscript more accessible to readers with varying levels of expertise.

2-A more detailed comparison between PTFOS grids and traditional ECoG grids, including quantitative data, could further strengthen the manuscript. Highlighting specific metrics where PTFOS outperforms traditional methods would be advantageous.

3-Discussing potential future directions for research and development in this area could provide a broader perspective on the long-term impact and ongoing advancements. This would help readers understand the trajectory of this technology and its applications.

4-Ensure consistency in terminology and formatting throughout the manuscript.

5-Address any grammatical errors or typos to improve the overall readability.

Author Response

Critique-1: For broader readability, it would be beneficial to include brief explanations of technical terms and concepts, such as inversion recovery (IR) sequence and steady state signal equation. This would make the manuscript more accessible to readers with varying levels of expertise.

Response-1: We thank the reviewer for their concern regarding the readability for a broader audience. Our main focus in this manuscript was to demonstrate that the new PTFOS implantation provides usable MR images, in contrast to the distorted images produced by traditional ECoG devices. The primary artifact observed with traditional ECoG protocols was the disturbance of magnetic field homogeneity induced by the ECoG devices, which significantly impacts the imaging quality of certain MRI sequences that are particularly sensitive to magnetic field susceptibility. To address this, we tested image quality using various clinical MRI sequences with different sensitivities to magnetic field homogeneity. However, explaining how susceptibility affects individual MRI sequences is complex and requires prior knowledge of MRI principles, which is beyond the scope of this manuscript. Therefore, we have modified our writing to include explanations for the MRI images and their sensitivity to the susceptibility artifacts. The following was added:

 The focus of this manuscript is to demonstrate that the new PTFOS implantation provides usable MR images, in contrast to the distorted images produced by traditional ECoG devices, which lead to disturbances in magnetic field homogeneity. We tested image quality with PTFOS implantation using various clinical MRI sequences with different sensitivities to magnetic field homogeneity, including T1-weighted (T1W), proton density (PD), T2-weighted (T2W), fast low-angle shot (FLASH), ultra-short TE (UTE), diffusion tensor imaging (DTI), and functional MRI (fMRI). The conventional spin-echo-based sequences, including T1W, PD, and T2W, are typically insensitive to magnetic field inhomogeneity, whereas the FLASH sequence, a steady-state conventional sequence, as well as EPI and DTI sequences, are typically sensitive to field inhomogeneity [6].”

Critique-2: A more detailed comparison between PTFOS grids and traditional ECoG grids, including quantitative data, could further strengthen the manuscript. Highlighting specific metrics where PTFOS outperforms traditional methods would be advantageous.

Response-2: Traditional ECoG for UH-MRI does not exist, so a comparison is not feasible. However, we have compared traditional ECoG grids and PTFOS for both MRI up to 7T and CT in our Radiology paper. The discussion now includes the following text:

“Our previous study showed that PTFOS induces no appreciable artifacts on CT and MR images when used over a human cadaveric head specimen [2]. For comparison, the head specimen was also imaged without a grid and with conventional grids. To quantify image quality, we enlisted board-certified neuroradiologists with decades of experience. They evaluated the images and rated the quality of brain tissue overlaid with PTFOS grids significantly higher than that with conventional ECoG grids (two-tailed t-test, P < 0.05). This study highlights the superior imaging compatibility of PTFOS grids, which do not compromise image quality or introduce artifacts. This is particularly important for clinical and research settings where accurate imaging is critical. The neuroradiologists' ratings underscore the potential of PTFOS grids to enhance diagnostic precision and effectiveness in neuroimaging applications.”

Critique-3: Discussing potential future directions for research and development in this area could provide a broader perspective on the long-term impact and ongoing advancements. This would help readers understand the trajectory of this technology and its applications.

Response-3: The future work will involve bridging with other Open Software projects, particularly the ephys. We have rewritten the future work paragraph as follows:

“We are focused on developing a plugin in C++ to seamlessly integrate continuous HF-2 data into the Open Ephys' existing signal chain. Open Ephys GUI is a pioneering and widely used application for multichannel electrophysiology that leverages a plugin-based workflow. This integration will allow users to utilize current plugins for various functionalities, including displaying the data, conducting real-time analysis, and saving it in multiple formats, such as the Neurodata Without Borders (NWB) standard. This integration process is designed for efficiency, ensuring full functionality without requiring additional development efforts. Open Ephys will provide seamless compatibility with the Data Portal "the DANDI" Archive, fulfilling NIH Data Management and Sharing Policy (DMSP) requirements. Furthermore, Open Ephys includes an open-source simulation management platform called StimJim, which will be compatible with HF-2. By developing this plugin, we aim to enhance the utility and versatility of HF-2 data, enabling researchers to take full advantage of the Open Ephys ecosystem for advanced electrophysiological studies.”

Critique-4: Ensure consistency in terminology and formatting throughout the manuscript.

Response-4: The paper has been carefully edited to ensure consistency and formatting.

Critique-5: Address any grammatical errors or typos to improve the overall readability.

Response-5: The paper has been carefully edited to ensure it is free from grammar mistakes and typos.

Round 2

Reviewer 2 Report

Comments and Suggestions for Authors

I am satisfied with the revision of the manuscript.

Reviewer 3 Report

Comments and Suggestions for Authors

The authors addressed all my comments.